# Chromosome Numbers and Genome Sizes of All 36 Duckweed Species (*Lemnaceae*)

**DOI:** 10.3390/plants11202674

**Published:** 2022-10-11

**Authors:** Phuong T. N. Hoang, Jörg Fuchs, Veit Schubert, Tram B. N. Tran, Ingo Schubert

**Affiliations:** Leibniz Institute of Plant Genetics and Crop Plant Research (IPK), Gatersleben, D-06466 Stadt Seeland, Germany

**Keywords:** chromosome number, duckweeds, evolution, genome size, karyotype, *Lemnaceae*

## Abstract

Usually, chromosome sets (karyotypes) and genome sizes are rather stable for distinct species and therefore of diagnostic value for taxonomy. In combination with (cyto)genomics, both features provide essential cues for genome evolution and phylogenetic relationship studies within and between taxa above the species level. We present for the first time a survey on chromosome counts and genome size measurement for one or more accessions from all 36 duckweed species and discuss the evolutionary impact and peculiarities of both parameters in duckweeds.

## 1. Introduction

A review about the aquatic monocot model family of duckweeds and its emerging economic importance has been published recently [1]. However, at this time, not for all duckweed species chromosome counts and genome sizes were known. Here, we compile corresponding data for at least one accession (clone) for each species, for some of them for the first time (see Figure 1, Results and Discussion). For previous data see [2] and references therein.

## 2. Material and Methods

The duckweed accessions investigated in this paper are listed in the legend of Figure 1 and in Figure 2 and were kindly provided by Dr. K.-J. Appenroth, Friedrich-Schiller-University, Jena, Germany. For plant cultivation, chromosome preparation, super-resolution microscopy and genome size measurement, see [2].

## 3. Results and Discussion

### 3.1. The Chromosomes

At least for one accession of all 36 duckweed species, chromosome counts are now available (for review see Figure 1 and [2]). For six species, *Lemna tenera* 9024 (2*n* = 40), *Wolffiella caudata* 9165 (2*n* = 42), *Wolffiella repanda* 9062 (2*n* = 40), *Wolffia cylindracea* 9056 (2*n* = 60), possibly a triploid accession, *Wolffia elongata* 9188 (2*n* = 40) and *Wolffia neglecta* 9149 (2*n* = 40), chromosome counts (Figure 2) are presented for the first time in this paper. Most frequent are chromosome counts around 2*n* = 40. However, in some papers [2,5,6] (and in older references in [2,4]), highly variable chromosome numbers were reported for the same species, e.g., for *Lemna aequinoctialis* 20, 40, 50, 60 and 80 [5]; 40, 50, 66, 72, 78, 84 and 65–76 [7]; 42, 84 [6]; 40, 60, 80 [4]; and 42 [2]. Despite the predominantly asexual propagation of many duckweed species, such a high variability is strange. Moreover, in cases where previously variable chromosome numbers were reported and the corresponding clones are still available, deviating numbers could not be confirmed [2]. In all tested clones of *Spirodela polyrhiza* (2*n* = 40) and *Spirodela intermedia* (2*n* = 36), we constantly found the same chromosome number, even for clones for which deviating numbers were reported before (see [2]). Nevertheless, smaller deviations in chromosome number between asexual clones of the same species cannot be excluded due to the potential presence of B chromosomes. Some of the samples apparently indicate autotetraploidy (*Le. aequinoctialis,* clone 2018 2*n* = 42, and clone 6746 2*n* = 84, refs. [2,6]) which occurred spontaneously, or was induced [8] as in *Landoltia punctata* with 2*n* = 46 and 2*n* = 92 (clone 5562 [2]). However, the quite high number of rather small chromosomes, which barely show any morphological differentiation (primary constrictions = centromeres, and secondary constrictions = nucleolus organizing regions [NOR], are mostly not discernible), and which are difficult to prepare from the small meristems, can lead to counting errors. For instance, for *Wolffiella rotunda* clone 9072 (2*n* = 82), fluorescent in situ hybridization (FISH), with 45S rDNA as a probe, revealed chromosome satellites, distally flanking the NOR, which could have easily been miscounted as small separate chromosomes [2] (Figure S3 therein). Furthermore, small and poorly structured chromosomes often make it difficult to decide whether they are mono- or holocentric. 

### 3.2. The Genome Sizes

There are several approaches to determining genome size, defined as pg/2C or Mbp/1C (C means the unreplicated haploid [or meiotically reduced] genome). Older papers applied Feulgen densitometry of nuclei, later ones mostly use flow cytometry [9] which allows for a much larger number of nuclei to be measured. These direct measurements require an internal reference standard with a known genome size close to that of the target species. Recently, based on sequenced genomes, k-mer distribution, as a bioinformatic approach, also served for genome size estimation. The accuracy of flow cytometry depends on the quality of the prepared nuclei suspension and the suitability of the used buffer and reference species. For precise genome size measurements, it is recommended that the coefficient of variation (CV) does not exceed 3% [9]. The k-mer size results provide a useful estimation, which should be complemented by direct measurements [10,11]. For all 36 duckweed species of the five genera, the genome size has been measured (Figure 1 and [2,3,4,12]). The values for *Lemna perpusilla* 8539 (519 Mbp), *Le. tenera* 9024 (526 Mbp), *Lemna turionifera* 8693 (475 Mbp), *We. caudata* 9165 (772 Mbp); *Wolffiella denticulata* 8221 (717 Mbp), *Wolffiella neotropica* 8848 (599 Mbp), *Wolffiella oblonga* 9391 (755 Mbp), 2*n* = 42), *We. repanda* 9062 (1190 Mbp, 2*n* = 40), *Wolffiella welwitschii* 9469 (780 Mbp), *Wo. cylindracea* 9056 (2144 Mbp; 2*n* = 60), *Wo. elongata* 9188 (936 Mbp; 2*n* = 40) and *Wo. neglecta* 9149 (1354 Mbp; 2*n* = 40) were obtained by flow-cytometry in this paper. The results yielded a ~14-fold range from 160 Mbp in the phylogenetically oldest genus, comprising *S. polyrhiza* and *S. intermedia,* up to 2203 Mbp of *Wolffia arrhiza* of the phylogenetically youngest genus [2]. The largest intrageneric variation was found in the genus *Wolffia* with 432 Mbp for *Wolffia australiana* to 2203 Mbp for *Wo. arrhiza.* For 11 species representing all duckweed genera, genome size was positively correlated with nuclear and cell volume of guard cells, and with progressive organ reduction, but negatively with frond size and phylogenetic age. No correlation was observed with the number of chromosomes or the rDNA loci [2]. The larger genomes of phylogenetically younger genera do not necessarily indicate a general evolutionary genome expansion in duckweeds. The high ratio of soloLTRs to intact retroelements of ~8 in *S. polyrhiza* [13], and the respective value of 11–14 in *Wo. australiana* [14]—the species with the smallest genome in its genus—could rather be a hint that the oldest genus *Spirodela*, after early branching from the other duckweeds, experienced genome shrinking due to a deletion-biased DNA double-strand break repair pathway [15]. The same could be true for *Wo. australiana* within its genus. In these cases, the likely scenario would be a bi-directional evolution of genome size starting from an intermediate ancestral duckweed genome of ~400 to 800 Mbp. Studies on DNA double-strand break repair in species with a high ratio of soloLTRs to intact retroelements could test this assumption. Genome size estimations for *S. polyrhiza*, *S. intermedia* and *La. punctata* from different laboratories are rather similar [2,3,4]. The other genomes measured by Hoang et al. [2] are in general larger than those measured for the same species by Wang et al. [4]. This might be caused by the use of different internal reference standards or different equipment, but most likely by assuming a different genome size of *Arabidopsis thaliana* as a basis for calculation (157 Mbp in [2], *versus* 147 Mbp in [4]). An initially unexpected finding was the strong variation of genome size estimation for clones of the same species. In case of clones of *Le. aequinoctialis* and *La. punctata,* the explanation is spontaneous, respectively induced, whole genome duplication (WGD) [2] (Figure 1). The genome sizes for two groups of *Le. minor* clones (~400 Mbp *versus* ~600 Mbp), however, differ by >50% ([4]; M. Bog, J. Fuchs, K.-J. Appenroth unpublished). For clone 5500, a genome size of 481 Mbp was estimated [16] and 409 Mbp for clone 8623 with 2*n* = 42 chromosomes [2]. For clone 8627, we counted 63 chromosomes and measured a genome size of >600 Mbp (PTNH & JF unpublished). Molecular fingerprinting for tubulin genes [17] suggested that clone 8627 belongs to *Lemna japonica* and is a hybrid of *Lemna minor* and *Le. turionifera.* Our data identified it as triploid. Further investigations are needed to uncover whether in other duckweed species with a high intraspecific variation of DNA content, clone-specific WGD with a fast subsequent genome size reduction occurred, or whether these groups of clones represent cryptic species or interspecific hybrids. The occurrence of interspecific hybrids in duckweeds is a surprise because they require sexual propagation—sometimes even involving unreduced gametes as in clone 8627 (see above)—which, according to laboratory observations, might occur very rarely in predominantly vegetatively propagating duckweed species. On the other hand, the maintenance of interspecific hybrids seems to be favored by asexual propagation.

### 3.3. Evolutionary Impact of Genome Size and Karyotype Studies

Chromosome numbers and genome size, varying in parallel by the same whole-number multiple, suggest a recent WGD (neopolyploidy) usually yielding autotetraploids in asexual clones. However, a nearly doubled genome size (or its absence) as well as a (nearly) doubled chromosome number (or its absence) alone are not sufficient to decide whether or not a WGD took place. Dysploid chromosome number reduction and/or fast reduction of genome size may blur real WGD. Therefore, in such cases, additional independent approaches are mandatory for arriving at conclusive statements. For instance, in Australian *Brassicaceae* with *n* = 4 to 6, evidence for mesopolyploidy (descendent dysploidy after WGD) was found through multicolor cross-FISH with bacterial artificial chromosome (BAC) pools representing the genome of *A. thaliana* [18]. All regions labeled by BAC pools from Arabidopsis, which represent the eight ancestral *Brassicaceae* chromosomes, appeared duplicated within the only four to six meiotic bivalents of the tested species. On the other hand, a genome can expand to double and more without WGD just by insertion-biased double-strand break repair (mainly retroelement insertion). In general, confirmed WGD leading to neopolyploidy seems to be more frequent between clones of distinct duckweed species than being responsible for genome size differences between species of the same or different genera. Of the natural species studied, only *We. rotunda* (clone 9072: 2*n* = 82; 1914 Mbp/1C), *Wo. arrhiza* (clone 8872: 2*n* = 60; 2203 Mbp/1C) and *Wo. cylindracea* (clone 9056: 2n = 60; 2144 Mbp/1C) are, so far, candidates for neopolyploids.

Karyotype studies employing molecular cytogenetics may help to elucidate the evolutionary origin of polyploid species. Genomic in situ hybridization (GISH) can identify the parental species if species-specific repetitive sequences are present in the genomes of the suspected parental species. For ‘quasi-diploid’ or paleopolyploid species (in fact all land plants experienced at least one very remote WGD), FISH with anchored unique sequences is an independent and direct approach to confirm or correct genome assemblies which are based on probabilistic methods. In this way, a robust chromosome-scale genome map has been achieved in several steps for *S. polyrhiza* [19,20,21]. Cross-FISH with single copy sequences can uncover chromosome homeology and chromosome rearrangements between related species. The two *Spirodela* species (*S. polyrhiza*, 2*n* = 40 and *S. intermedia,* 2*n* = 36) were studied by sequential multicolor cross-FISH with different pools of 96 BACs anchored in the genome of *S. polyrhiza* [22] and compared with genome assemblies for both species [21]. Eight chromosome pairs did not reveal rearrangements between both species. The other twelve chromosome pairs of *S. polyrhiza* correspond to the remaining ten pairs of *S. intermedia* and display (in part multiple) rearrangements (Figure 3). The direction of the evolution cannot be determined with certainty, because the genus comprises only these two species. However, because most duckweed species tend to have 40 or more chromosomes, 40 might be the ancestral and 36 the derived chromosome number. No rearrangements were found between seven clones from different geographic origins of *S. polyrhiza* [20] and two clones of *S. intermedia* [21], respectively. These studies showed clearly that *S. polyrhiza* and *S. intermedia* are different species, despite similar genome size and overlap of most morphological features. In contrast to the situation described for the *Brassicaceae* family (e.g., ref. [23]), cross-FISH with *S. polyrhiza* BACs yielded only weak and dispersed signals but did not recognize chromosome homeology when applied to species of other duckweed genera (*La. punctata*, *Le. aequinoctialis*, *Wolffiella hyalina*, *Wo. arrhiza*), independently of stringency conditions [24]. So far, cross-FISH with oligo probes derived from chromosome assemblies of *S. polyrhiza* or *S. intermedia* also did not reveal chromosome homeology in other duckweed genera [25]. The probes yielded either dispersed signals in *La. punctata* and *Le. aequinoctialis* or no signals at all as in *We. hyalina* and *Wo. australiana.* Apparently, these genomes are too diverse and probe densities are too low to generate reliable signals across the duckweed genera. In future, homologous single copy sequences, selected assuming synteny between the genomes in question, should be designed for oligo-FISH to unassembled genomes [25]. 

## 4. Conclusions

Having now compiled chromosome counts and genome sizes for at least one accession of all 36 duckweed species (for some species these are the first data), a basis is available upon which to investigate the evolution of duckweed genomes as well as the reasons for the apparent intraspecific variation of both parameters. A future combination of genomic data, genome size, chromosome number, GISH and FISH data will clarify the phylogenetic position and taxonomic status of intrageneric duckweed accessions which are difficult to assign to distinct species on the basis of morphological criteria.

## Figures and Tables

**Figure 1 plants-11-02674-f001:**
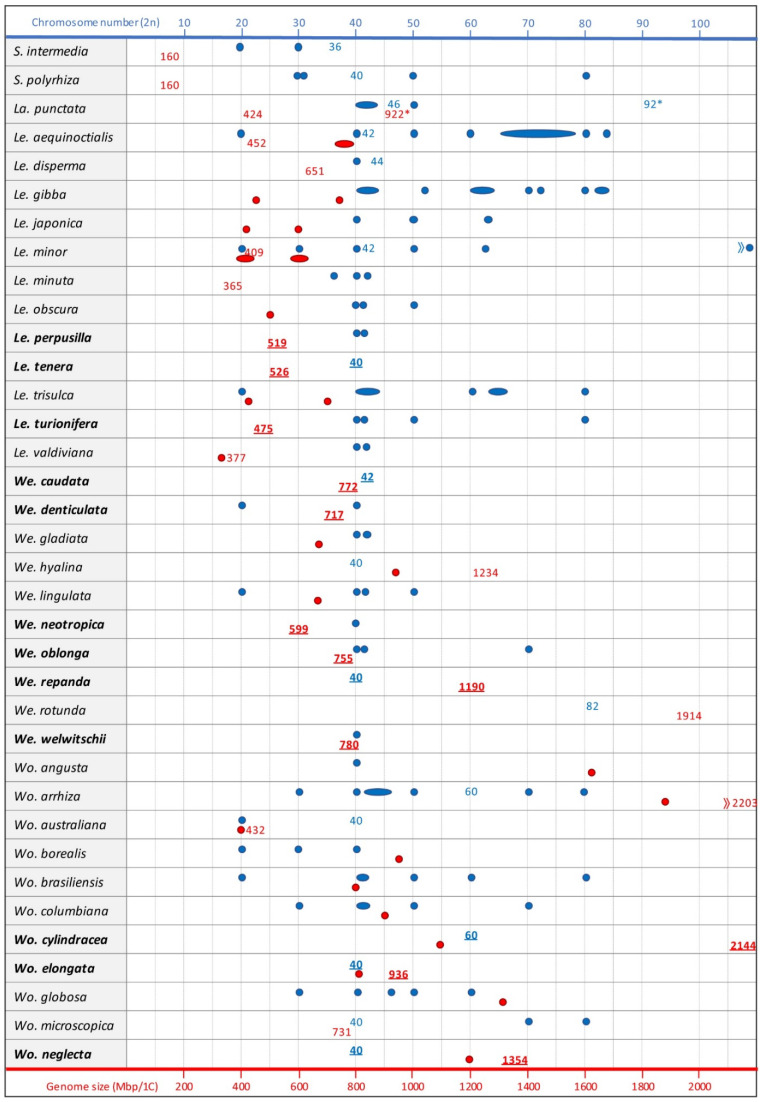
Chromosome number (blue) and genome size (red) of duckweeds. When chromosome numbers and genome sizes refer to newly investigated species or were taken from [2] or [3] (*Le. minuta*), they appear as numbers. For references to other chromosome counts (blue dots and ovals [for closely adjacent values]) see [2]. Other genome size estimations (red dots and ovals) were from [4]. The values for *Le. perpusilla* 8539 (519 Mbp), *Le. tenera* 9024 (526 Mbp), *Le. turionifera* 8693 (475 Mbp), *We. caudata* 9165 (772 Mbp); *We. denticulata* 8221 (717 Mbp), *We. neotropica* 8848 (599 Mbp), *We. oblonga* 9391 (755 Mbp), 2*n* = 42), *We. repanda* 9062 (1190 Mbp, 2*n* = 40), *We. welwitschii* 9469 (780 Mbp), *Wo. cylindracea* 9056 (2144 Mbp; 2*n* = 60), *Wo. elongata* 9188 (936 Mbp; 2*n* = 40), *Wo. neglecta* 9149 (1354 Mbp; 2*n* = 40) were obtained by flow-cytometry and determined in this paper. Species for which we provide in this paper the first or new values are in bold and the corresponding values are underlined. * indicates the values for the colchicine-induced tetraploid *La. punctata* clone 5562.

**Figure 2 plants-11-02674-f002:**
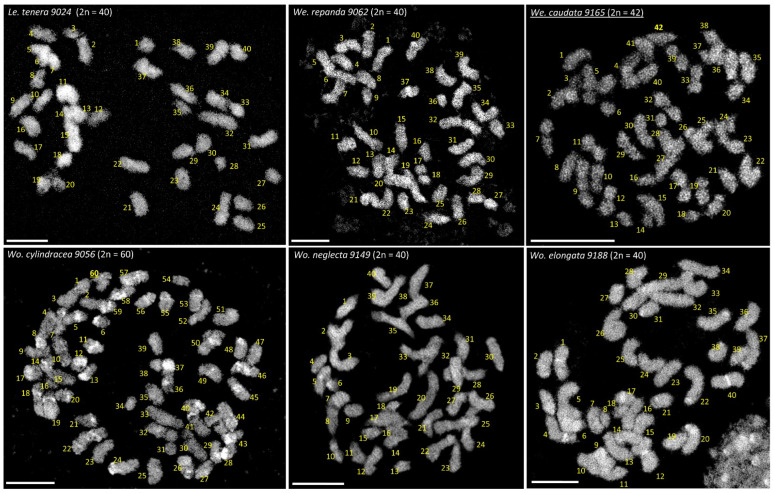
Metaphases with chromosome counts reported for the first time in this paper. To count clearly the chromosome number, image stacks from spatial super-resolution structured illumination microscopy (3D-SIM) [3] were used to identify overlapping chromosomes. Individual chromosomes were enumerated arbitrarily. Scale bar: 5 µm.

**Figure 3 plants-11-02674-f003:**
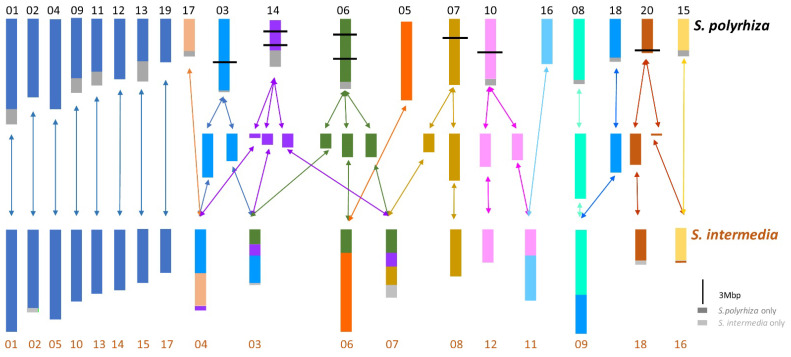
Chromosome homeology and rearrangements between *S. polyrhiza* (*n* = 20) and *S. intermedia* (*n* = 18) as revealed by cytogenomics, see [21,22]. The double arrows indicate that either direction of evolution could have happened. Dark blue are chromosomes not involved in rearrangements; other colors indicate rearranged chromosomes. Dark grey boxes: regions present only in *S. polyrhiza*; light grey boxes: regions present only in *S. intermedia.* Enumeration indicates distinct chromosomes of the respective species.

## Data Availability

Not applicable.

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
