# Peer review of "Chromosome Numbers and Genome Sizes of All 36 Duckweed Species (Lemnaceae)"

_plants, 2022, doi:10.3390/plants11202674_

Round 1

Reviewer 1 Report

This manuscript provides interesting scenario. However, its presentation is not acceptable because it does not allow differentiating what is known and what is not known. It is written like a note and the author's need to re-evaluate its presentation. In its current format I do not accept it to be published. 

The authors need to revisit the presentation and make it to be in line with how manuscripts in Plants are published. For instance:  I would like to see an Introduction, Materials & Methods, Results, and Discussion. 

Author Response

"However, its presentation is not acceptable because it does not allow differentiating what is known and what is not known."

This statement of reviewer 1 needs specification, otherwise we do not know what is requested.

"It is written like a note and the author's need to re-evaluate its presentation. In its current format I do not accept it to be published. The authors need to revisit the presentation and make it to be in line with how manuscripts in Plants are published. For instance:  I would like to see an Introduction, Materials & Methods, Results, and Discussion. "

"(x) Extensive editing of English language and style required "

May be, one or the other sentence in our manuscript is not elegant enough, then please specify. However, in none of the >300 scientific papers I have authored or co-authored so far, I received a request of this type during the reviewing process. 

Reviewer 2 Report

This short review by Phuong et al. summarizes years of reports related to genome size and chromosome number evaluations in the duckweed family. New data for the few missing species are also added, completing the picture for the 36 extant duckweed species. This is a hot topic in the field, starting to reveal the complexity of duckweed phylogenesis and evolution.

The manuscript is exhaustive and well written. Some points, listed below, need minor revision or answers.

Lines 19-20: I don’t think this reference needs to be cited in the Abstract

Lines 36-37: please justify this sentence, is it supported by any reference?

Lines 37-38: please specify the clone number as in the case of Landoltia, for uniformity and better clarity

Line 39: as in La. punctata

Line 48: In order to enhance new data, chromosome counts from newly analyzed clones (Fig 2) should be reported and commented here rather than in the legend of Figure 2 (see below). Interpretation for chromosome counts deviating from the “standard” value of 40-42 (Wo. cylindracea) could also be provided

Line 62: Authors should mention here, instead of in the legend of Fig1, the genome size of the newly measured clones which allowed completing Fig 1 with all species.

Line 72 and 79: Please, use the same terminology to indicate solo LTR/intact retroelements ratio

Lines 81-82: this sentence could sound misleading since variation in genome size are found between clones of the same species. Please rephrase to avoid misunderstanding

Lines 84-85: since Authors already described variations in chromosome numbers for the same species, this shouldn’t be completely unexpected, but seems in line with chromosome number variation

Lines 88-89: please also refer to Fig1

Line 91: PTNH & JF??

Line 92: clone number is 8627, not 8726. Ref 17, describing this clone as a hybrid, does not suggest it is triploid, rather that this clone and others reported as L. minor are in fact L. japonica.

Line 94: species instead of clones, I think.

Line 122-124: Why not including also Wo. cylindracea (2n = 60)?

Line 124: Why do Authors say neotetraploid species? On the basis of what reported before in the text, could triploidy or disploidy be excluded? A more generic polyploid would fit better. In addition, since high intraspecific variability is reported for many species, it is important to specify that genome size measurement and chromosome counting for a single clone may not be sufficient to infer the genome size and ploidy at the species level. In the case of Wo. arrhiza many clones were investigated with different chromosome counts (Fig 1), while for We. rotunda just one clone was analyzed. Please also add the reference for the two mentioned clones.

Figure 1

In the Figure legend, for some clones there is inconsistency between chromosome counts reported here and in Figure 2 (please check We. caudata vs. We oblonga and Le. tenera)

As mentioned before, I suggest that the list of the newly analyzed clones (chromosome numbers and genome size measurements) should be better reported in the text, in the dedicated chapters, rather than in the figure legend. In addition, chromosome counts of the six new clones are already reported in Figure 2 and its legend.

Line 163: Figure legend must indicate that also these new data (chromosome counts and genome size measurements) are indicated by numbers in Figure 1. New data could be highlighted in Figure 1, e.g. by using bold style, in order to highlight them.

Line 164: what’s the difference between dots and ovals? Are ovals confluent dots from many clones? Please specify

Lines 164-165: since Ref 2 and 4 also report many older data, please specify if also these data are reported in the figure or not

Lines 171-172: This sentence related to Spirodela is rather a comment than a description of the Figure and should be moved to the main text.

Figure2: the clone number for Le. tenera is missing in the legend. Please also add scale bar measure in the Figure or in the legend

Lines 180-181: please clarify what you mean by “as exemplified for We. caudata and Wo. cylindracea” or indicate by arrows

Line 186: regions instead of sequences seems preferable

Author Response

Response to reviewers

Many thanks to this reviewer for constructive suggestions.

"Lines 19-20: I don’t think this reference needs to be cited in the Abstract"

We have removed this reference from the Abstract into a short Introduction.

"Lines 36-37: please justify this sentence, is it supported by any reference?"

We inserted now into this sentence 'due to the potential presence of B chromosomes'.

"Lines 37-38: please specify the clone number as in the case of Landoltia, for uniformity and better clarity"; clone numbers are inserted now.

"Line 39: as in La. punctata" 'as' is inserted now.

"Line 48: In order to enhance new data, chromosome counts from newly analyzed clones (Fig 2) should be reported and commented here rather than in the legend of Figure 2 (see below). Interpretation for chromosome counts deviating from the “standard” value of 40-42 (Wo. cylindracea) could also be provided" Done on p. 2.

"Line 62: Authors should mention here, instead of in the legend of Fig1, the genome size of the newly measured clones which allowed completing Fig 1 with all species." Done on p.3.

"Line 72 and 79: Please, use the same terminology to indicate solo LTR/intact retroelements ratio" Done

"Lines 81-82: this sentence could sound misleading since variation in genome size are found between clones of the same species. Please rephrase to avoid misunderstanding" Done by insertion of 'in general'.

"Lines 84-85: since Authors already described variations in chromosome numbers for the same species, this shouldn’t be completely unexpected, but seems in line with chromosome number variation" We wrote now 'An initially unexpected finding...'

"Lines 88-89: please also refer to Fig1" Done

"Line 91: PTNH & JF??"

The authorship for this future paper is not yet clear, but definitely the three mentioned names will be included.

"Line 92: clone number is 8627, not 8726. Ref 17, describing this clone as a hybrid, does not suggest it is triploid, rather that this clone and others reported as L. minor are in fact L. japonica." Corrections done (thank you for detecting transposed digits).

"Line 94: species instead of clones, I think." Corrected.

"Line 122-124: Why not including also Wo. cylindracea (2n = 60)?"

We modified this sentence correspondingly.

"Line 124: Why do Authors say neotetraploid species? On the basis of what reported before in the text, could triploidy or disploidy be excluded? A more generic polyploid would fit better." Done, see above.

"In addition, since high intraspecific variability is reported for many species, it is important to specify that genome size measurement and chromosome counting for a single clone may not be sufficient to infer the genome size and ploidy at the species level. In the case of Wo. arrhiza many clones were investigated with different chromosome counts (Fig 1), while for We. rotunda just one clone was analyzed. Please also add the reference for the two mentioned clones."

The values for We. rotunda (9072) were reported by Hoang et al. (2019)=[2]. For Wo. arrhiza we counted 60 chromosomes in clone 8872, other counts were originally from Urbanska (1980) and Geber (1989) c.f. Hoang et al. (2019)=[2].

"Figure 1

In the Figure legend, for some clones there is inconsistency between chromosome counts reported here and in Figure 2 (please check We. caudata vs. We oblonga and Le. tenera)"

We could not find inconsistency for the three species mentioned (We. oblonga is not in Fig. 2 because counts were from Geber (1989) and Urbanska (1980).)

"As mentioned before, I suggest that the list of the newly analyzed clones (chromosome numbers and genome size measurements) should be better reported in the text, in the dedicated chapters, rather than in the figure legend. In addition, chromosome counts of the six new clones are already reported in Figure 2 and its legend." Done, see above.

"Line 163: Figure legend must indicate that also these new data (chromosome counts and genome size measurements) are indicated by numbers in Figure 1." Done. 

"New data could be highlighted in Figure 1, e.g. by using bold style, in order to highlight them."

Species for which we provide in this paper the first or new values are now in bold and the corresponding values are underlined.

"Line 164: what’s the difference between dots and ovals? Are ovals confluent dots from many clones? Please specify" Done.

"Lines 164-165: since Ref 2 and 4 also report many older data, please specify if also these data are reported in the figure or not" Yes, as said (dots and ovals).

"Lines 171-172: This sentence related to Spirodela is rather a comment than a description of the Figure and should be moved to the main text." Done on p. 3.

"Figure2: the clone number for Le. tenera is missing in the legend. Please also add scale bar measure in the Figure or in the legend"

Le. tenera 9024 is now inserted on p. 2; the scale bar size in the legend.

"Lines 180-181: please clarify what you mean by “as exemplified for We. caudata and Wo. cylindracea” or indicate by arrows"  

This was a remnant of a previous manuscript version, which had to be deleted.

"Line 186: regions instead of sequences seems preferable" Done.
